# Antimicrobial Resistance and Comparative Genomic Analysis of *Elizabethkingia anophelis* subsp. *endophytica* Isolated from Raw Milk

**DOI:** 10.3390/antibiotics11050648

**Published:** 2022-05-12

**Authors:** Pavel A. Andriyanov, Pavel A. Zhurilov, Daria D. Kashina, Anastasia I. Tutrina, Elena A. Liskova, Irina V. Razheva, Denis V. Kolbasov, Svetlana A. Ermolaeva

**Affiliations:** 1Branch in Nizhny Novgorod, Federal Research Center for Virology and Microbiology, 603950 Nizhny Novgorod, Russia; zhurilov95@bk.ru (P.A.Z.); dasha.kashina99@mail.ru (D.D.K.); nastya.tutrina@yandex.ru (A.I.T.); liskovaea@mail.ru (E.A.L.); razheva64@bk.ru (I.V.R.); drermolaeva@mail.ru (S.A.E.); 2Federal Research Center for Virology and Microbiology, 601125 Volginsky, Russia; kolbasovdenis@gmail.com

**Keywords:** *Elizabethkingia*, *Elizabethkingia anophelis* subsp. *endophytica*, comparative genomics, drug resistance, pathogenicity

## Abstract

*Elizabethkingia anophelis* is an emerging multidrug-resistant pathogen that causes severe nosocomial and community-acquired infections worldwide. We report the first case of *E. anophelis* isolation in Russia and the first isolation from raw cow’s milk. The ML-44 demonstrated resistance to 28 antimicrobials of 33 tested in the disk-diffusion test. Whole genome-based phylogeny showed ML-44 strain clustered together with the F3201 strain isolated from a human patient in Kuwait in 1982. Both strains were a part of the “*endophytica*” clade. Another clade was formed by subsp. *anophelis* strains. Each of the *E. anophelis* compared genomes carried 18 to 21 antibiotic resistance determinants. The ML-44 chromosome harbored nine efflux system genes and three beta-lactamase genes, along with six other antimicrobial resistance genes. In total, 72 virulence genes were revealed. The set of virulence factors was quite similar between different *E. anophelis* strains and included LPS and capsule encoded genes, type IV pili, oxidative stress response genes, and genes encoding TIVSS and TVISS effectors. The particular interest caused the *mip* and *zmp*1 gene homologs, which can be essential for intracellular survival. In sum, our findings suggest that raw milk might be a source of *E. anophelis* harboring a set of virulence factors and a broad resistance to generally used antimicrobials.

## 1. Introduction

*Elizabethkingia anophelis* is a Gram-negative, non-motile, oxidase-, and catalase-positive aerobic bacilli [1]. Recently, it has emerged as an opportunistic pathogen that causes life-threatening nosocomial and community-acquired infections and outbreaks [2]. *E. anophelis* infection has high mortality rates and manifests by sepsis, meningitis, nosocomial pneumonia in immunocompromised and immunocompetent persons [3]. Outbreaks of nosocomial *E. anophelis* infection were registered in the Central African Republic, Singapore, Hong Kong, England, and Taiwan in 2011–2013 [4,5,6,7,8]. In 2016, an unusually large community-associated outbreak involving 65 persons and leading to 20 fatal cases was registered in the Midwestern United States (Wisconsin, Illinois, and Michigan) [9]. Several sporadic cases were also reported from Denmark and France [10,11]. To date, *E. anophelis* has not been reported from Russia.

*E. anophelis* infection is complicated by the intrinsic multidrug resistance of the pathogen. *E. anophelis* carries multiple chromosomally encoded determinants that provide resistance to the majority of beta-lactam and aminoglycoside antibiotics. Mainly, *E. anophelis* is a unique bacterium that possesses three chromosomally encoded beta-lactamase genes: *blaB*, *blaCME*, and *blaGOB* [12,13,14]. The spectrum of antibiotics effective against *E. anophelis* is restricted to minocycline, doxycycline, piperacillin–tazobactam combination, fluoroquinolones, trimethoprim–sulfamethoxazole, and rifampin [9,15,16,17].

Based on the whole genome phylogenetic analysis, *E. anophelis* species was divided into two subtaxons: subsp. *anophelis* and subsp. *endophytica* [18]. The first one is more frequently isolated from different sources, including human patients and clinical environments, and has confirmed pathogenicity to humans. The second one was first isolated from healthy internal stem tissues of sweet corn (*Zea mays*) and established as a novel species of *E. endophytica*, considered a plant endophyte [19]. In 2016, it was reassessed as *E. anophelis* subsp. *endophytica* [18]. To date, only a few cases of *E. anophelis* subsp. *endophytica* isolation have been reported. Besides the type strain JM-87, isolated from corn, the F3201 strain was isolated from a human host in Kuwait in 1982. The pathogenic potential of *E. anophelis* subsp. *endophytica* is still elusive.

In this study, we reported the *E. anophelis* subsp. *endophytica* strain, isolated from raw milk in Russia. We applied phenotype-based methods to assess physiological capabilities and antimicrobial susceptibility and whole-genome sequencing and comparative genomics approach to investigate genome features and phylogeny and determined similarities and differences with other *E. anophelis* strains.

## 2. Results

### 2.1. Isolation and Phenotypic Characterization of the E. anophelis Subsp. endophytica ML-44 Strain

We isolated *E. anophelis* in monitoring studies from the raw milk sample obtained from the farm at the territory of the NN region, Russia. A single colony was taken from Endo agar as a sample of lactose non-fermenting bacteria contaminating raw milk. In addition to the E. anophelis, *Pseudomonas aeruginosa* and *Enterobacter aerogenes* were isolated from this milk specimen. Identification was based on the nearly full-length 16S rRNA gene sequence using 27f and 1492r primers. The 16S RNA consensus was 1348 bp in length with 99.71% sequence identity to the sequence of *E. anophelis* subsp. *endophytica* JM-87 (NR_136481.1). The culture was designated as the *E. anophelis* subsp. *endophytica* strain ML-44.

ML-44 formed milk-white colonies of 1–2 mm in diameter with a smooth surface on Endo agar. On tryptic soy agar, the isolate formed visible yellowish colonies approximately 3–4 mm in diameter within 24 h at 37 °C. Microscopic studies showed small non-motile rods with typical Gram-negative staining (Appendix A). Biochemical properties were obtained through ENTEROtest 24 N kit. ML-44 was oxidase, catalase, ß-galactosidase, and esculin positive and hydrolyzed trehalose and mannitol. The following tested activities were negative: urease, arginine hydrolase, ornithine decarboxylase, lysine decarboxylase, hydrogen sulfide activities, Simmons citrate, and malonate utilization, ß-xylosidase activity. No acid was detected from salicine, sorbitol, melibiose, cellobiose, lactose, dulcitol, adonitol, arabitol, sucrose, inositol, raffinose (Appendix A).

### 2.2. E. anophelis ML-44 Possesses an Alpha-Hemolytic Activity

Hemolytic activity is typical for many pathogenic bacteria, but it has not been described in *E. anophelis* subsp. *endophytica* yet. *E. anophelis* ML-44 did not exhibit yellow pigment on the blood agar; colonies were smooth and white. After 48 h of incubation, ML-44 exhibited zones of α-hemolysis. The hemolysis zones were more apparent on the agar with rabbit blood. Agar zones were more visible on the sheep blood when bacterial biomass was removed (Figure 1).

### 2.3. E. anophelis ML-44 Showed a Multidrug-Resistant Phenotype

*E.**anophelis* ML-44 drug susceptibility was determined using a standard disc-diffusion method to 33 antimicrobials. Antibiotics tested, and the size of inhibition zones are listed in Table 1. ML-44 isolate was resistant to 28 antibiotics used in the clinical practice, including penicillin G, ampicillin, amoxicillin-clavulanate, aztreonam, ticarcillin, piperacillin, piperacillin-tazobactam, ticarcillin-clavulanate, ceftazidime, cefotaxime, cefepime, imipenem and meropenem, kanamycin, neomycin, tobramycin, gentamicin, amikacin, polymyxin, clindamycin, clarithromycin, erythromycin, tylosin, nalidixic acid, trimethoprim, trimethoprim-sulfamethoxazole, chloramphenicol, and tetracycline. The strain was susceptible to rifampicin, ciprofloxacin, levofloxacin, enrofloxacin, and linezolid.

### 2.4. Genomic Traits of E. anophelis ML-44

The assembly of the ML-44 genome contained 31 contigs with 4.03 M bp genome size and 35.4% average GC-content. In total, 3805 features were detected in the ML-44 chromosome; among them, 3758 CDS and 47 RNAs were predicted. CDS distribution among specific subsystems was predicted using the RAST web tool. Two hundred sixty subsystems were prescribed. The major subsystems were “Amino Acids and Derivatives” (264 CDSs), “Carbohydrates” (131 CDSs), “Cofactors, Vitamins, Prosthetic Groups, Pigments” (128 CDSs), and “Protein Metabolism” (126 CDSs). Notably, 36 and 11 CDSs were assigned as “Virulence, Disease and Defense” and “Phages, Prophages, Transposable elements, Plasmids” categories. “Virulence, Disease, and Defense” subsystem contained 24 CDSs related to antimicrobial and toxic compounds resistance as well as 12 CDSs involved in the invasion process and intracellular resistance.2.5. Whole Genome-Based Phylogeny and Comparative Analysis.

We used a whole genome-based phylogeny approach realized via the REALPHY web tool to determine a position of ML-44 among 14 complete sequenced *E. anophelis* strains (Figure 2). The ML-44 strain clustered together with the strain F3201 isolated from the human host in Kuwait in 1982. Both strains were a part of the common “*endophytica*” clade and formed an individual subcluster distinct from the subcluster that included the type JM-87 strain isolated from corn. Inside the “*anophelis*” clade, different strains formed a few subclades, including the separated CSID_3015182678-CSID_3015182681 subclade associated with the Wisconsin outbreak of 2016.

Additionally, we performed OrthoANI analysis to confirm the intraspecies level of the ML-44 strain (Figure 3). The ANI value for the ML-44-F3201 pair was the highest among all other *E. anophelis* strains. These findings were consistent with the results of the REALPHY phylogeny analysis. Genome-to-genome distance evaluation showed that only members of the *endophytica* phylogroup had a DDH value of more than 90% with the ML-44 genome (Table 2), and other strains belonged to *anophelis* phylogroup have DDH < 80. Based on the Genome-to-Genome Distance Calculator results, the *E. anophelis* ML-44 strain had the highest similarity with the JM-87 (93.5%) and F3201 (93.2%). In addition, none of the strains had a DDH level of less than 70%, indicating that all of them belong to the *E*. *anophelis* species.

### 2.5. Comparison of Coding Sequences

We compared the strain ML-44 with the *E. anophelis* strains F3201, OSUVM2, JM-87 (subsp. *endophytica*), and R26, CSID_3015183678 (subsp. *anophelis*) using the OrthoVenn2 web platform. Protein sequences formed 4074 clusters, 1090 orthologous clusters (at least two species), and 2984 single-copy clusters. In total, 3019 clusters, including 18,207 proteins, were analyzed (Figure 4). Strains F3201 and OSUVM-2 held the highest number of singletons, 308 and 263, respectively, whereas ML-44 and JM-87 had the lowest number of singletons, 145 and 213, respectively. Based on OrthoVenn2 statistical analysis, the ML-44 genome formed the most numerous cluster (3681) with R26 type strain and the smallest one with CSID_3015183678 Wisconsin strain (3654). Additionally, the latter genome demonstrated the smallest proteome amongst the studied ones. Proteomes of ML-44 and OSUVM2 were quantitatively similar, 3597 and 3544 clusters, respectively.

### 2.6. Comparative Resistome Analysis

We used whole-genome sequencing data to reveal a genetic basement of the ML-44 multi-drug resistance and compared genetic determinants of antibiotic resistance in ML-44 and other *E. anophelis* strains. According to the CARD database, genomes of 7 *E. anophelis* strains (ML-44, F3201, OSUVM2, JM-87, CSID_3015183678, NUHP2, R26) contain at least 29 distinct AMR genes divided into three general groups according to resistance mechanisms: factors associated with antibiotic efflux, antibiotic inactivation, and antibiotic target alteration (Appendix A). Individual strains carried from 18 to 21 AMR determinants. All strains shared nine of 14 detected efflux systems, while antibiotic inactivation genes were strain-specific. The ML-44 strain carried 18 AMR genes, including 11 genes related to the antibiotic efflux, seven genes associated with antibiotic inactivation, and one gene with antibiotic target alteration. The detected AMR determinants appeared to provide ML-44 resistance to beta-lactams, aminoglycosides, macrolides, tetracycline, fluoroquinolones, and even disinfecting agents intercalating dyes resistance. Four different beta-lactamase genes were found in the ML-44 genome, including *bla*B-18, *bla*CME -12, *bla*GOB-41, and *csp*-1. The first three genes are specific to the *Elizabethkingia* genus and confer resistance to penams, cephalosporins, carbapenems. CSP-1 enzyme was initially isolated from *Capnocytophaga sputigena* in 2010. It is a metallo-beta-lactamase that confers resistance to amoxicillin, ticarcillin, narrow-spectrum cephalosporins, ceftazidime, cefotaxime, and aztreonam [20]. Aminoglycoside resistance in the ML-44 strain might be realized via the *aadS* gene and the *ranA-ranB* ABC efflux system genes [21]. Tetracycline resistance might be realized via homologs of the efflux systems *txR* and *adeF*. Multiple homologs of the efflux pump genes, including *abeS*, *arlR*, *ceoB*, *macB*, *mefS*, might be responsible for phenotypic resistance to macrolides. The ML-44 genome included efflux genes *adeF*, *arlR*, and *ceoB* that were described as conferring resistance to fluoroquinolones, while the strain did not exhibit phenotypic resistance to any of the fluoroquinolones tested. Both non-functionality of the efflux systems under conditions tested and the unsuitability of parameters used for phenotypic resistance definition could be responsible for this discrepancy. Taken together, genomic analysis data demonstrated that chromosomally encoded determinants provide the ML-44 eminent phenotypic multidrug resistance, and a noticeable part of these determinants seems to be shared by the majority of a strain belonging to the species *E. anophelis*.

### 2.7. Virulome Analysis

In sum, 92 homologs of known virulence factors (VF) were predicted among seven analyzed *E. anophelis* genomes (Appendix A). Most of them were related to the outer membrane (LPS, O-antigen, and LOS) and capsule formation (genes of exopolysaccharide and alginate synthesis). The following VF categories were also detected: adhesion factors, including type IV pili, heme uptake, degradation, synthesis factors, LPS synthesis factors, metal ion transport proteins, stress response and stress survival factors, and several others. The ML-44 strain carried a restricted set of capsule synthesis enzymes and factors involved in the LPS synthesis that requires further studies. Preliminary results did not reveal a capsule in ML-44 (data not shown). Some virulence-associated factors recognized in *E. anophelis*, including heme uptake systems and anti-oxidant enzymes catalase and superoxide dismutase, are characteristic of many pathogenic bacteria. The hemolytic assay demonstrated hemolytic activity in the strain ML-44. Still, the *hly* gene encoding α-Hemolysin in the *E. anophelis* type strain JM-87 was absent from ML-44.

Among more specific virulence factors, the macrophage infectivity potentiator (Mip) homolog gene was found in all *E. anophelis* strains. The Mip protein that has been firstly found in Legionella pneumophila is required for macrophage infection and survival in freshwater protozoa [22,23]. A few effector proteins of the Dot-Icm type IVB secretion system (TIVSS) were found in the M-44 genome (Appendix A) that could be another feature shared by *E. anophelis* and *L. pneumophila*. Still, we failed to find the structural TIVSS genes in the *E. anophelis* genomes. Several regulatory and auxiliary proteins of the type VI secretion system (TVISS) suggested that *E. anophelis* strains could carry this secretion system, too. Another enzyme that could be assigned to factors responsible for intracellular survival in ML-44 is a zinc metalloprotease 1 (Zmp1) homolog. Zmp1 is a *Mycobacterium tuberculosis* secreted peptidase that mediates key stages of tuberculosis disease progression [24]. Taken together, obtained data suggested that *E. anophelis* possesses some determinants used by intracellular parasites for infection in humans.

### 2.8. Prophage and CRISPR Comparative Investigation

Prophage regions with different completeness levels were detected among seven compared *E. anophelis* genomes. Only the strain F3201 harbored a complete prophage with 49.8 Kb sequence length, which contained 46 encoded proteins and an incomplete phage of 7.6 Kb in length. In ML-44 strain, two incomplete phages were found: 14.5 Kb and 8.3 Kb with 16 and 9 CDS. Observed CDS were annotated such as phage-like proteins, protease, fiber protein, head protein, and tail protein, but no integrase was detected among all phage CDS in the ML-44 genome. Other strains contained incomplete (score < 70) prophages with different phage protein genes except for integrases. Genomes of OSUVM2, JM-87, CSID_3015182678, NUHP2, and R26 harbored 3, 3, 1, 6, and 1 prophage regions, respectively.

In F3201, OSUVM2, CSID_3015182678, NUHP2, and R26 strains, the CRISPR-Cas system elements were not detected. Only ML-44 and JM-87 genomes had CRISPR-Cas elements. JM-87 strain had two repeats with a length of 45 bp and one spacer region, but Cas genes were not indicated. Interestingly, the ML-44 chromosome possessed a class IIC CRISPR-Cas system. Fifteen direct repeats, each with a length of 47 bp and 14 spacer regions with 30 bp in length, were identified in the ML-44 CRISPR-Cas system. In addition, this system had a complete set of Cas genes: Cas9, Cas1, and Cas2 (Appendix A).

## 3. Discussion

*E. anophelis* is a recognized emerging pathogen that causes nosocomial severe and community-acquired infections among immunocompromised and immunocompetent persons globally. In addition to the severe disease infection and high mortality, this pathogen is characterized by multidrug resistance, increasing its importance for clinic practice [9,10,25,26].

Here, we report the first case of isolation of *E. anophelis* subsp. *endophytica* in Russia. Besides its novelty from the point of view of geographic distribution, the obtained isolate has other features that make it unique. Firstly, the reported *E. anophelis* subsp. *endophytica* strain ML-44 possessed hemolytic activity toward rabbit and sheep erythrocytes. To the best of our knowledge, this is the first *E. anophelis* subsp. *endophytica* strain with the hemolytic activity registered in vitro. Another essential characteristic of the strain is its food origin. *E. anophelis* seems to be a gut commensal of *Aedes* and *Anopheles* mosquitoes [27,28,29]. Still, there is no evidence that mosquitoes serve as a vector to transmit the bacteria to humans [9]. Isolation of the pathogen was reported from diverse environments such as hospital tap waters and sink, environmental aquatic environments, human clinical specimens, corn, and horses [4,7,19,25,30]. These data suggested that *E. anophelis* is a widespread environmental pathogen. We speculate that *E. anophelis* can resemble in its ecology such well-established environmental pathogens as *Pseudomonas aeruginosa*, *Legionella pneumophila*, *Listeria monocytogenes*, *Yersinia pseudotuberculosis*, etc. This class of pathogens is sometimes named soilborne or saprozoic human pathogens because their evolution is associated with natural soil or water ecosystems [31,32,33]. Environmental pathogens are usually characterized by polytonality, i.e., an ability to colonize a wide range of hosts, including humans, wild and domestic animals, plants, and protozoa [34,35]. Environmental pathogens often occur in farm environments, food plants, and food products [36,37].

The reported isolation of the *E. anophelis* strain ML-44 from raw milk aligns with the hypothesis about its belonging to the diverse group of environmental pathogens. We can only guess whether milk contamination occurred due to the dairy cow′s infection or if it occurred during milk transportation to the market. Both routes of milk contamination are possible for soilborne human and animal pathogens. There are no data that *E. anophelis* can cause disease or persist in cows. Still, in 2016, William L. Johnson et al. [30] reported the isolation of two *E. anophelis* strains from equine specimens. Phylogenetic analysis of the core genome revealed a high similarity of these equine isolates with clinical isolates that can reflect the probability of transmission of this agent among human and animal populations. Interestingly, both equine isolates formed one phylogenetic clade with the JM-87 type strain of *endophytica* subspecies, F3201 isolate of clinical origin, and our ML-44 strain isolated from the raw milk sample in Russia.

Obtained results demonstrated that all tested *E. anophelis* strains have a similar range of virulence factors (VFs) despite strains belonging to different subspecies, *E. anophelis* subsp. *anophelis* and subsp. *endophytica*. Notably, along with previously observed *E. anophelis* VF genes such as *katG*, *IlpA*, *clpP*, *rmlA*, *htpB*, *DnaK*, etc., which can be implicated in the invasion, defense, and persistence [26], we also found that all stains in our study harbor macrophage infectivity potentiator (*mip*) homolog gene. The Mip protein was shown to facilitate the establishment of the *L. pneumophila* intracellular infection cycle in free-living protozoa and macrophages [22,38]. Another virulence factor found in ML-44 and other strains belonging to both *E. anophelis* subspecies is a homolog of the *M. tuberculosis* metalloprotease Zmp1 that can provide macrophages survival. The presence of these factors and proteins of the TIVSS suggests that *E. anophelis* may have an intracellular stage during the infection process in humans and is in line with the hypothesis about *E. anophelis* as an environmental pathogen that can survive in protists during ecological existence. Taken together, obtained data demonstrated that *E. anophelis* belonging to both species shared a VF suggesting that both subspecies might have a similar pathogenic potential in humans. Still, these findings should be extrapolated with caution because mentioned potential virulence factors have not been studied in detail.

*E. anophelis* has been previously known to be highly resistant to many antibacterials. This phenomenon is caused by the possession of many resistance genes, including different antibiotic degrading enzymes and efflux pumps [30]. Our results are in line with previous findings. We found resistance to penicillins, cephalosporins, monobactam, and carbapenems. This is consistent with the fact that the ML-44 genome carries multiple resistance determinants. Four different beta-lactamase genes were found in the ML-44 genome, such as blaB-18, blaCME-12, blaGOB-41, and csp-1. The first three are specific to the *Elizabethkingia* genus and confer resistance to penams, cephalosporins, carbapenems. Resistance to carbapenems is of particular concern because the latter is frequently used in empirical antimicrobial therapy in acute infection cases caused by Gram-negative bacilli [39]. CSP-1 enzyme is a metallo-beta-lactamase initially isolated from *Capnocytophaga sputigena* in 2010. It is known to confer resistance to amoxicillin, ticarcillin, narrow-spectrum cephalosporins, ceftazidime, cefotaxime, and aztreonam in *E. coli* TOP10 recombinant strain [20].

There was no drug to which our strain was sensitive among five tested aminoglycoside antimicrobials. The presence of the *aadS* gene probably caused aminoglycoside resistance in the ML-44 strain. Interestingly, the latter gene is silent in wild-type Bacteroides, and its expression is dependent on trans-acting chromosomal mutation. The protein has significant homology to streptomycin-dependent adenyltransferase of Gram-positive bacteria [40]. Additionally, such resistance may be caused by the presence of the RanA/RanB ABC efflux homologous system. It was previously shown that this efflux confers resistance to streptomycin and amikacin and organic solvents such as dimethyl sulfoxide and some alcohols. Notably, this system is also involved in virulence in *R. anatipestifer* [21]. Tetracycline resistance in ML-44 can be mediated by the presence of *adeF*, *tetB*(46), and *txR* predicted gene homologs [41,42,43]. Particular proteins are components of efflux machinery that play a specific role in conferring tetracycline resistance. Other genes associated with efflux systems such as *abeS*, *arlR*, *ceoB*, *macB* identified in the ML-44 chromosome can be implicated in macrolide, phenicol, pleuromutilin, aminoglycoside, and even disinfecting agents and intercalating dyes resistance.

According to the disk diffusion assay, the *E. anophelis* ML-44 strain had no fluoroquinolone resistance phenotype. The most frequently occurring resistance mechanism to the latter one is a mutation in the DNA gyrase A enzyme subunit. Previously Lin et al. [17] have shown a significantly increased level of levofloxacin and ciprofloxacin MICs induced by Ser83Ile, Ser83Arg, or Ala709Ser substitutions in *gyrA* subunit of 11 clinical isolates of *E. anophelis*. We aligned ML-44 *gyrA* with another gyrase A subunits from different *E. anophelis* strains, including EM361-97 strain, which has Ser83Ile substitution and increased MIC to fluoroquinolones [44]. ML-44 carries Ser at 83 position, which confirms the susceptible phenotypic profile of our strain to fluoroquinolones. Notably, strains that formed *anophelis* and *endophytica* clades have a kind of reversion in amino acid sequence at 841–842 positions related to the hydrophobicity of particular positions (Appendix A). Members of *anophelis* clade have Ala841 and Ile842, whereas *endophytica* clade members like ML-44 have Val841 and Ala842. These findings also support phylogenetic differences between these two groups. Moreover, reversions of this kind can affect the MIC values of fluoroquinolones. However, this hypothesis lacks validity, and more experiments are needed to check this assumption.

Despite the found correlation between phenotypic resistance and resistance genes possessed in the ML-44 chromosome, we recognize the limitations of this study. Nowadays, criteria for *E. anophelis* antimicrobial susceptibility determination via disk diffusion approach are absent. Antibiotics such as ticarcillin-clavulanate, piperacillin, piperacillin-tazobactam, and cefepime, which had a zone inhibition diameter other than zero in our study, can be used to prevent infection and not be interpreted with confidence. According to Chiu C.-T. et al. [45], the disk diffusion method and Etest are not appropriate for AST assays for *E. anophelis* susceptibility estimating due to the high rate of errors. Only minocycline, rifampin, levofloxacin, and ceftazidime can be used in AST via the disk diffusion approach. Nonetheless, the disk diffusion method is still the most convenient, inexpensive, and widely used AST method. Criteria for antimicrobial susceptibility testing of *E. anophelis* as well as a clear treatment scheme are required because the mentioned microbe has high pathogenicity and ultimate resistance potential and can cause severe infections in humans. Thus, our data will be helpful for the AST criteria development for the *E. anophelis* disk diffusion approach.

Overall, obtained data demonstrated that the raw milk might be a source of *E. anophelis* strains possessing a set of virulence factors and a notable resistance to generally used antimicrobials. Further monitoring of this pathogen might be helpful to prevent its establishment as a widely spread nosocomial pathogen.

## 4. Materials and Methods

### 4.1. E. anophelis Isolation and Identification

The *E. anophelis* strain ML-44 was isolated via Endo agar medium (HiMedia) in raw milk monitoring studies. An unpasteurized milk sample came from a farm in the Nizhny Novgorod region of Russia. The sample was serially diluted and aseptically plated on Endo agar. After 24 h, a single colony with a smooth glittering surface and white-colored was detected. For further research, a colony was streaked onto tryptic soy agar (HiMedia).

Strain identification was performed through 16s rRNA gene sequencing as follows. A single colony was suspended in 100 µL of sterile deionized water and lysed by incubation at 95 °C for 10 min. PCR reaction was carried out with the universal bacterial primers specific for 16s rRNA gene: 27F: 5′-AGAGTTTGATCMTGGCTCAG-3′ and 1492R: 5′-TACGGYTACCTTGTTACGACTT-3′ [46]. PCR product was extracted and purified from the agarose gel to sequence using the ABI PRISM BigDye Terminator v. 3.1 reagent kit (Applied Biosystems) according to the manufacturer’s protocols, followed by an analysis of the reaction products and an automatic sequencer Applied Biosystems 3730 DNA Analyzer. The 16S rRNA nucleotide sequences were assembled using the Unipro UGENE (v39.0) software [47]. The obtained consensus was compared to other bacterial 16s rRNA genes with the EzTaxon server database [48].

### 4.2. Physiological and Biochemical Characterization of E. anophelis ML-44

Biochemical features were determined with the ENTEROtest 24 N kit (Erba Lachema). OXItest (Erba Lachema) was used to detect oxidase activity. The standard biochemical test with 3% hydrogen peroxide was used to detect catalase activity.

### 4.3. Antimicrobial Susceptibility Testing

Susceptibility to 33 antimicrobials was tested using the disk diffusion method by EUCAST 2021 (European Committee on Antimicrobial Susceptibility Testing) guidance. The following antibiotics were tested: kanamycin (30 µg), neomycin (30 µg), gentamicin (10 µg), tobramycin (10 µg), amikacin (30 µg), amoxicillin-clavulanate (20–10 µg), ampicillin (10 µg), aztreonam (30 µg), piperacillin-tazobactam (30–6 µg), ticarcillin (75 µg), piperacillin (30 µg), ticarcillin-clavulanate (75–10 µg), ceftazidime (10 µg), cefotaxime (5 µg), cefepime (30 µg), imipenem (10 µg), meropenem (10 µg), penicillin G (1 unit), clindamycin (2 µg), clarithromycin (15 µg), erythromycin (15 µg), tylosin (15 µg), levofloxacin (5 µg), enrofloxacin (5 µg), ciprofloxacin (5 µg), nalidixic acid (30 µg), trimetoprim (5 µg), rifampicin (5 µg), linezolid (30 µg), chloramphenicol (30 µg), tetracycline (30 µg), trimetoprim-sulfametoxazol (1.25–23.75 µg), and polymyxin (300 U). Susceptibility breakpoints for Pseudomonas aeruginosa were extrapolated for most antimicrobials, due to the lack of breakpoints for *Elizabethkingia* species. For macrolides interpretation (clarithromycin, erythromycin, tylosin) we used standard developed for *Campylobacter jejuni*; for clindamycin, linezolid we used *Enterococcus* spp. breakpoints; for rifampicin, nalidixic acid, and trimethoprim-sulfamethoxazole we used *Haemophilus influenzae* breakpoints, and *Yersinia enterocolitica* breakpoints were used for tetracycline. Mueller–Hinton agar and most antibiotic-containing disks were obtained from HiMedia; amoxicillin-clavulanate, gentamicin, polymyxin, trimethoprim-sulfamethoxazole were obtained from NICF; aztreonam, piperacillin-tazobactam, ticarcillin, piperacillin, ticarcillin-clavulanate, tobramycin, amikacin were obtained from Bioanalyse.

### 4.4. Hemolysis Assays

We used Columbia blood agar base (HiMedia) to evaluate hemolytic activity with 5% (v-v) rabbit blood and blood agar with 5% sheep blood. Overnight culture of the *E. anophelis* ML-44 strain was streaked onto the surface of the agar and incubated for 48 h at 37 °C. The positive reaction was interpreted as a clear zone of lysis of the red blood cell (beta-hemolysis) or color-changing (alpha-hemolysis). The absence of hemolytic activity was interpreted as gamma-hemolysis.

### 4.5. Whole-Genome Sequencing, Assembly, and Annotation

According to the manufacturer′s instructions, genomic DNA was extracted from overnight *E. anophelis* culture using the QIAamp DNA Kit (Qiagen, Düsseldorf, Germany). DNA concentration was estimated using the Qubit dsDNA BR Assay Kit (Fisher Scientific, Waltham, MA, USA) on a Qubit 3.0 fluorometer (Fisher Scientific). Obtained DNA sample was sequenced using an Illumina HiSeq 1500 (Illumina, San Diego, CA, USA) by 150-bp paired-end reads with coverage of 200x at Geneanalytics LLC. Reads were trimmed via Trimmomatic software (v0.4.8) with adapter removal [49]. Assembly was performed with Unicycler v0.4.8. with trimming option (Trim Galore v0.4.2) and polishing option (2 rounds of Pilon v1.23) on the PATRIC resource center (https://patricbrc.org/ (accessed on 7 October 2021).) [50]. Eventually 4,034,074 bp assembly consisting of 31 contigs with N50 = 297,287 and L50 = 5 was received. Annotation was carried out using the Rapid Annotations Subsystems Technology (RAST) server [51,52,53] and the Prokaryote Genome Annotation Pipeline. The genome sequence has been deposited at GenBank (JAJNCD000000000).

### 4.6. Comparative Genomic Analysis of E. anophelis Strains

A phylogenetic tree based on whole-genome sequence comparison was reconstructed using the reference sequence alignment-based phylogeny builder (REALPHY v1.13) with default parameters [54]. In sum, 14 genomes of strains with different origins were included in the phylogenetic analysis; among them, 7 are clinically pathogenic strains or human-associated, 3 are animal associated (2 with horse and 1 with fish), 3 have an environmental origin, and ML-44 strain had a dairy product (raw milk) source (Table 3). Only seven genomes were complete. All genomes were obtained from the NCBI database (www.ncbi.nlm.nih.gov) (accessed on 20 December 2021).

We assembled the panel of 7 *E. anophelis* genomes for a further comparative approach. Panel included the genome of our ML-44 strain and six genomes from the GenBank database: *E. anophelis* F3201, OSUVM2, JM87, CSID_3015182678, NUHP2, R26. Strains in the panel had a different origin from environmental to clinical.

The OrthoANI values calculation with closely related strains was performed using the Orthologous Average Nucleotide Identity Software Tool (OAT) with default parameters [55]. The heat map was visualized in TBtools [56]. Digital DNA–DNA hybridization (DDH) was calculated at the Genome-to-Genome Distance Calculator (GGDC 3.0) webserver [57]. The assembly of the ML-44 strain was hybridized against 20 publicly available genomes of *E. anophelis* strains of different origins. We used Formula 2 to assess DNA-DNA hybridization values (DDH), as the latter is independent of genome length and can be efficiently utilized for incomplete draft genomes. We also carried out orthologous clustering analysis with the OrthoVenn2 web server with default parameters, but E-value was set up as ≤1 × 10^−5^ [58]. FASTA files containing protein sequences of 6 *E. anophelis* strains were used to predict clusters of orthologous genes: ML-44, JM-87, OSUVM2, F3201, and R26 CSID_3015182678.

Virulence factors (VFs) and antimicrobial resistance determinants were predicted in the panel of *E. anophelis* strains with BLASTP (v2.12.0) search against the virulence factor database (VFDB) set B [59] and the Comprehensive Antibiotic Resistance Database (CARD) [60] databases, respectively. We used seven strains of *E. anophelis* to compare genes mentioned above; the following strains were used: ML-44, F3201, OSUVM2, JM-87, CSID_3015182678, NUHP2, and R26. The cut-off parameters were set up as follows: Bitscore > 100, E-value ≤ 1 × 10^−30^, Identity > 40%, Query coverage > 50. Mentioned parameters were used in both searches (VDFB and CARD). Genes of the *gyrA* subunit were aligned using UGENE (v39.0) [47] with the ClustalW algorithm (v2.0) [61].

Prophage elements in the genome of ML-44 isolate were predicted by the PHASTER web server [62]. It implements BLAST search against custom bacteriophage databases then combines all matches in three subgroups based on score: intact (score > 90), questionable (score from 70 to 90), and incomplete (score < 70). To observe CRISPR elements in the ML-44 genome, we utilized the CRISPRCasFinder web server [63].

## Figures and Tables

**Figure 1 antibiotics-11-00648-f001:**
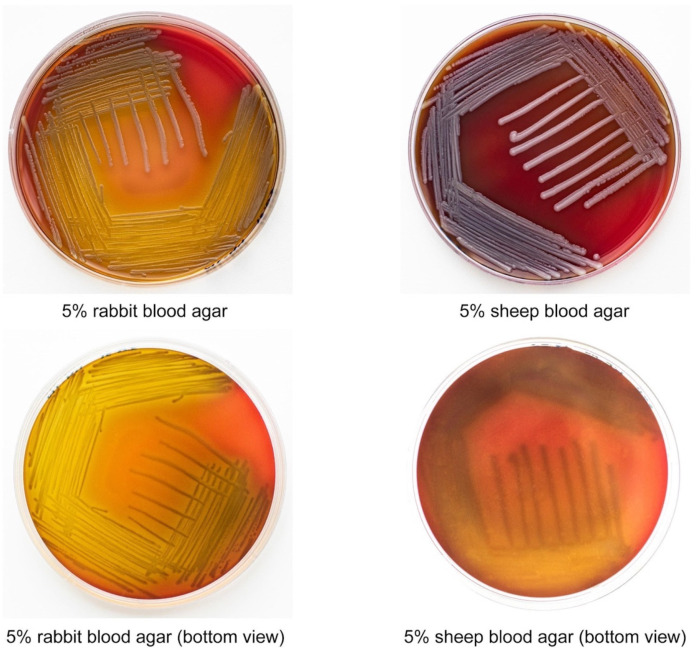
Elizabethkingia anophelis ML-44 on 5% rabbit blood agar and 5% sheep blood agar after 48 h of incubation.

**Figure 2 antibiotics-11-00648-f002:**
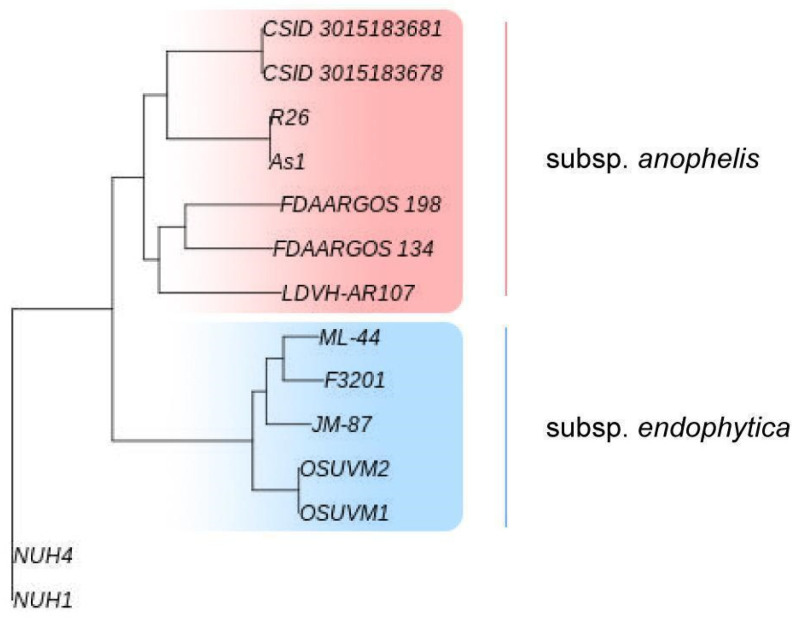
Reconstruction of the whole-genome tree formed by 14 *Elizabethkingia anophelis* strains. The subsp. *anophelis* clade is highlighted in red and the subsp. *endophytica* clade is highlighted in blue.

**Figure 3 antibiotics-11-00648-f003:**
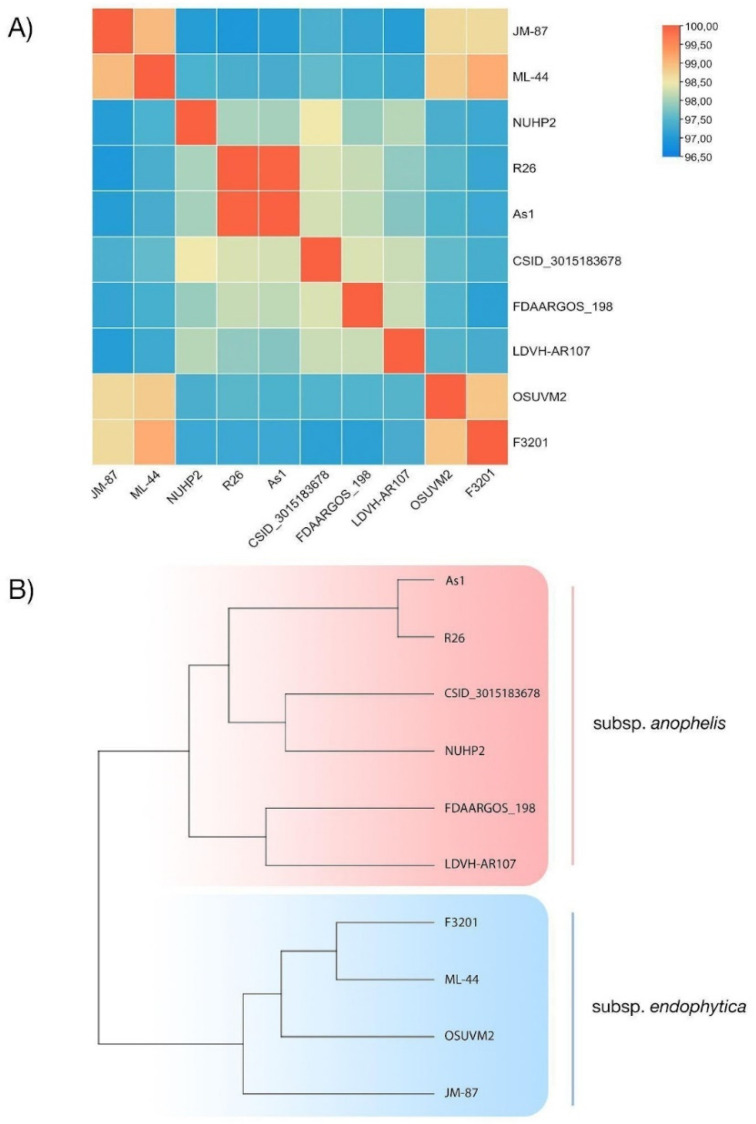
OrthoANI analysis results: (**A**) Heat map based on ANI values among ML-44 strain and other 9 *E. anophelis* strains; (**B**) Cladogram based on OrthoANI calculations.

**Figure 4 antibiotics-11-00648-f004:**
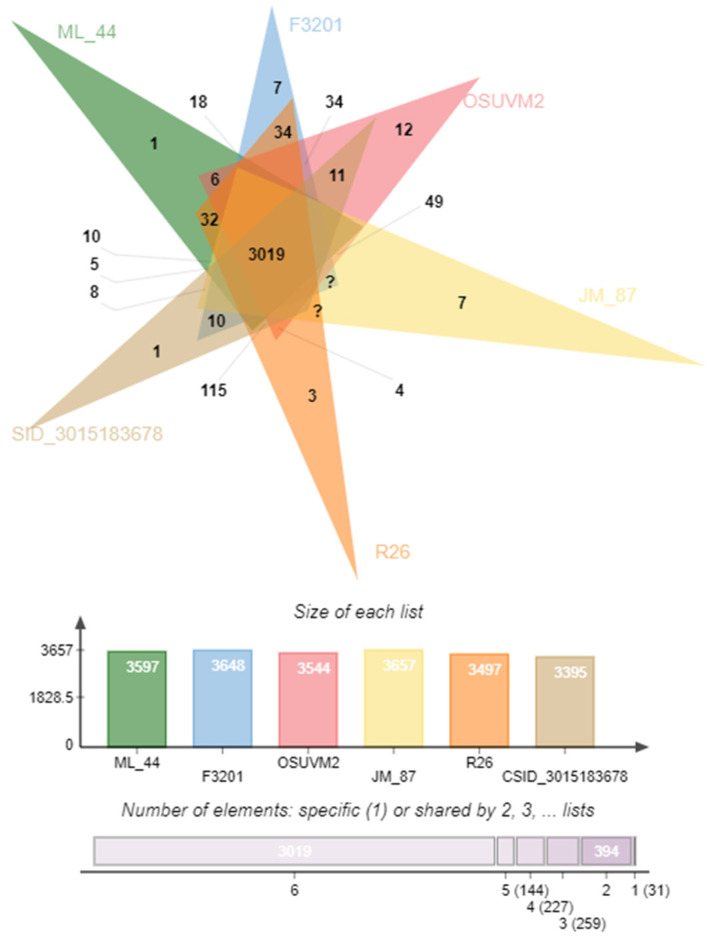
Results of ortholog genes analysis using OrthoVenn2. The Venn diagram shows the number of shared and unique orthologous genes clusters (gene families) among 6 *E. anophelis* strains.

**Table 1 antibiotics-11-00648-t001:** List of antibiotics tested and values of inhibition zones diameter.

Class	Antibiotic	Interpretation	Zone Diameter, mm
Aminoglycosides	Amikacin	R	0
	Gentamicin	R	0
	Kanamycin	R	0
	Neomycin	R	0
	Tobramycin	R	0
ß-lactams: Penicillins	Penicillin	R	0
	Ampicillin	R	0
	Ticarcillin	R	0
	Piperacillin	R	14
	Amoxicillin-Clavulanate	R	0
	Ticarcillin-Clavulanate	R	13
	Piperacillin-Tazobactam	R	16
ß-lactams: Cephalosporins	Ceftazidime	R	0
	Cefotaxime	R	0
	Cefepime	R	10
ß-lactams: Carbapenems	Imipenem	R	0
	Meropenem	R	0
ß-lactams: Monobactam	Aztreonam	R	0
Fluoroquinolones	Ciprofloxacin	S	32
	Levofloxacin	S	30
	Enrofloxacin	I	28
	Nalidixic acid	R	18
Macrolides	Clarithromycin	R	20
	Erythromycin	R	19
	Tylosin	R	13
Other	Linezolid	S	22
	Rifampicin	S	20
	Tetracycline	R	18
	Trimethoprim	R	13
	Trimethoprim-Sulfametoxazol	R	0
	Chloramphenicol	R	0
	Clindamycin	R	17
	Polymyxin	R	0

**Table 2 antibiotics-11-00648-t002:** Genome-to-genome distance calculator (GGDC) analysis for ML-44 strain and other related *E. anophelis* strains.

Query Genome	Reference Genome	DDH (f2), %
ML-44	JM-87	93.5
ML-44	F3201	93.2
ML-44	OSUVM2	90.5
ML-44	FDAARGOS_198	79.7
ML-44	CSID_3015183678	79.3
ML-44	NUHP2	78.3
ML-44	LDVH-AR107	78
ML-44	R26	77.3
ML-44	As1	77

**Table 3 antibiotics-11-00648-t003:** *Elizabethkingia anophelis* strains used in this study.

№	Strain Name	Source	Region/Country	Collection Date	WGS Status	WGS GenBank Accession No.
1	As1	Mosquito (*A. gambiae*)	Pennsylvania, USA	2013	Draft	LFKT01
2	CSID_3015183678	Human patient	Wisconsin, USA	2016	Complete genome	CP014805.2
3	CSID_3015183681	Human patient	Wisconsin, USA	2016	Complete genome	CP015068.2
4	F3201	Human host	Kuwait	1982	Complete genome	CP016374.1
5	FDAARGOS 134	Human patient	Washington, D.C., USA	2014	Complete genome	CP014021.1
6	FDAARGOS 198	Human patient	Sweden	Missing	Complete genome	CP023010.2
7	JM-87	Sweet corn (*Zea mays*)	Alabama, USA	2011	Complete genome	CP016372.1
8	LDVH-AR107	Common carp (*Cyprinus carpio*)	Montpellier, France	2004	Draft	FTPG01
9	NUH1	Human patient	Singapore	2012	Draft	ASYH01
10	NUH4	Human patient	Singapore	2012	Draft	ASYI01
11	NUHP2	Human patient	Singapore	2012	Draft	ASYF01
12	OSUVM1	Equine Stall	Oklahoma, USA	2016	Draft	PJMA01
13	OSUVM2	Horse (*Equus caballus*)	Oklahoma, USA	2016	Draft	PJLZ01
14	R26	Mosquito (*A. gambiae*)	Stockholm, Sweden	2005	Complete genome	CP023401.1

## Data Availability

Whole Genome Shotgun project has been deposited at DDBJ/ENA/GenBank under the accession JAJNCD000000000. BioProject (PRJNA783500), BioSample (SAMN23435286).

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
