# Peer review of "Antimicrobial Resistance and Comparative Genomic Analysis of Elizabethkingia anophelis subsp. endophytica Isolated from Raw Milk"

_antibiotics, 2022, doi:10.3390/antibiotics11050648_

Round 1
Reviewer 1 Report
Andriyanov et al. investigated Elizabethkingia anophelis, an emerging multidrug-resistant pathogen that causes nosocomial and community-acquired infections worldwide. Reporting the first case of E. anophelis in Russia isolation from raw cow's milk. Monitoring the antimicrobial resistance profile of strains isolated from a human patient in Kuwait through genomic analysis of microorganisms. The present manuscript collaborates directly with public health. The materials and methods are robust and the results support the discussion. This reviewer accepts the manuscript as it is and only suggests a text revision to facilitate reading. It would be interesting for the authors to insert a photo of the morphology of the microorganism (Gram) to help health professionals identify the microorganism with simple tools such as staining and biochemistry. This reviewer wishes the research group success and congratulates the valuable findings.
Reviewer 2 Report
The manuscript entitled “Antimicrobial resistance and comparative genomic analysis of Elizabethkingia anophelis subsp. endophytica isolated from raw milk” by Andriyanov et al. performed both experimental and bioinformatics studies to explore the virulence gene and antibiotic resistance potential of E. anopheles. This seems an interesting finding that can be published in Antibiotics journal. My specific comment is mentioned below.
Author used Blastp to predict the virulence and antibiotic resistance genes. However, it seems they used a less stringent cutoff (>40% identity and >50 query coverage) for their blastp search and hence could increase the probability of getting false positive prediction. Can author provide the rationale behind choosing less string cutoff for determining antibiotic resistance genes?
Reviewer 3 Report
In this manuscript the authors present the first Elizabethkingia anophelis subsp endophytica isolated from raw milk in Russia. The study is well designed and the results are clearly presented.
I have only one comments:
lines 308-314: it would be informative the analysis of the spacers in order to investigate if the spacers match phage genome
minor comment:
line 68: write Pseudomonas aeruginosa and Enterobacter aerogenes by italics
